# Structural Insights into the Dynamic Assembly of a YFV sNS1 Tetramer

**DOI:** 10.3390/v16081212

**Published:** 2024-07-29

**Authors:** Qi Pan, Qiang Chen, Wanqin Zhang, Haizhan Jiao, Lei Yu, Hongli Hu

**Affiliations:** 1Kobilka Institute of Innovative Drug Discovery, School of Medicine, The Chinese University of Hong Kong, Shenzhen 518172, China; 221059033@link.cuhk.edu.cn (Q.P.); chenqiang@cuhk.edu.cn (Q.C.); 118010418@link.cuhk.edu.cn (W.Z.); jiaohaizhan@cuhk.edu.cn (H.J.); 2Guangzhou Eighth People’s Hospital, Guangzhou Medical University, Guangzhou 510060, China

**Keywords:** YFV, sNS1, tetramer, dynamic assembly

## Abstract

Yellow fever virus (YFV) infections can cause severe diseases in humans, resulting in mass casualties in Africa and the Americas each year. Secretory NS1 (sNS1) is thought to be used as a diagnostic marker of flavivirus infections, playing an essential role in the flavivirus life cycle, but little is known about the composition and structure of YFV sNS1. Here, we present that the recombinant YFV sNS1 exists in a heterogeneous mixture of oligomerizations, predominantly in the tetrameric form. The cryoEM structures show that the YFV tetramer of sNS1 is stacked by the hydrophobic interaction between β-roll domains and greasy fingers. According to the 3D variability analysis, the tetramer is in a semi-stable state that may contain multiple conformations with dynamic changes. We believe that our study provides critical insights into the oligomerization of NS1 and will aid the development of NS1-based diagnoses and therapies.

## 1. Introduction

Yellow fever virus (YFV) is a serious mosquito-borne virus endemic to tropical areas of Central and South America, as well as sub-Saharan Africa [1]. In humans, YFV infection causes a series of clinical syndromes, including a viral hemorrhagic fever called yellow fever (YF) with a case fatality rate ranging from 20% to 60% [2]. Although the live-attenuated YFV vaccine (YFV-17D) is an effective weapon to prevent the disease, low vaccination coverage in endemic areas is difficult to overcome [3]. Since 2016, numerous outbreaks occurred in Brazil and other tropical regions, so there is an urgent need to develop new diagnostic and therapeutic methods [4,5,6].

YFV is a member of the *Flavivirus genus* of the *Flaviviridae family*, which also includes the Dengue virus (DENV), Zika virus (ZIKV), West Nile virus (WNV), Japanese encephalitis virus (JEV), and tick-borne encephalitis virus (TBEV) [7]. Flaviviridae viruses share a common genome structure consisting of a single-stranded and positive-sense RNA, which is approximately 11 kb length and encodes three structural proteins (Core, PrM, and E) and seven non-structural (NS) proteins (NS1, NS2A-2B, NS3, NS4A-B, and NS5) [8]. As the *multifunctional* protein within the Flaviviridae family, Non-structural protein 1 (NS1) is considered as a promising target against flaviviruses. The NS1 gene encodes a 48 to 55 kD glycoprotein that is well conserved among flaviviruses, exhibiting 20–40% identity and 60–80% similarity at the amino acid level [9]. Extracellularly secreted NS1 (sNS1) has been used as a serological diagnostic marker for dengue and is correlated with the severity of dengue disease [10]. Moreover, the sNS1 lipoprotein is involved in virion production, immune evasion, and multiple aspects of pathogenesis [11,12,13,14]. YFV sNS1s have been reported to induce endothelial dysfunction in vitro and in animal models [12], and, recently, Harris Group showed that YF disease severity is also associated with increased serum levels of the viral sNS1 [15].

Although the structural features of NS1 dimers have been extensively studied up to 2014, including the high-resolution crystal structures of the NS1 dimer of DENV [16], WNV [17] and ZIKV [18] and the C-terminal fragment of NS1 (NS1c) of YFV [19] and JEV [20], the composition and structure of higher oligomeric form of sNS1 is still poorly understood. Traditionally, the structures of sNS1 have been assumed to be homogeneously hexameric [10,21,22]. However, a recent study reported the DENV2 sNS1 from a clinical DENV2 strain PVP94/07 exists predominantly in the tetrameric form and has two distinct tetrameric states: a stable and a loose conformation [23]. In addition, the sNS1 dimer appears to be able to bind to human high-density lipoprotein (HDL) [24,25,26]. Syzdykova et al. also suggested that GFP-NS1 fusion proteins extracted from the supernatant of YFV-infected cells may contain tetramers and hexamers [27]. Although flavivirus NS1 exhibits 60–80% similarity at the amino acid level [9], the amino acid sequences of the β-roll domain responsible for its tetrameric assembly are not highly conserved. Therefore, there may be differences in the assembly of flavivirus sNS1 oligomers. Structural and functional studies targeting YFV NS1 may help us to understand the pathogenesis and develop diagnostic and therapeutic drugs against YFV [15,28].

Here, we observed that the recombinant sNS1 dimer from a YFV 17D strain exists in multiple-oligomeric states, predominantly in the tetrameric form. The cryoEM structure of the YFV sNS1 tetramer is determined at a global resolution of 3.57 Å and is further improved to 3.31 Å by focusing the refinement on one of the dimers. The YFV tetramer of sNS1 is stacked from two dimers by the hydrophobic interaction between β-roll domains and greasy fingers. According to the 3D variability analysis, the tetramer is in a semi-stable state that is highly flexible with dynamic conformations.

## 2. Materials and Methods

### 2.1. Cell Culture

Expi293F suspension cells [Union-Biotech (Shanghai) Co. Ltd.] (Shanghai, China) were cultured in 293 Expression medium (Union-Biotech (Shanghai) Co., Ltd.) with 1% penicillin/streptomycin at 37 °C and 5% CO_2_ and were maintained at 0.5–2.0 × 10^6^ cells/mL on a cell shaker at 110 rpm. This cell line was used to produce recombinant expressed YFV sNS1.

### 2.2. Construction, Expression, and Purification of YFV NS1

The pCAGGS expression vector consisting of YFV NS1 gene was a gift from Wei Zhao (Southern Medical University). The coding sequences of YFV were cloned into the pCAGGS expression vector with the N-terminal cd33 signal peptide (cite) and a C-terminal 6 × histidine tag. Transfections were performed using Expi293F suspension cells and linear polyethylenimine (PEI MAX^®^, Polysciences, Warrington, PA, USA) according to the manufacturer’s protocol, and supernatants containing YFV NS1 were collected 48 h post-transfection. The 6xHis-tagged recombinant YFV NS1 proteins were purified by Ni-NTA affinity resin and washed with 20 mM sodium phosphate, pH = 7.4, 150 mM NaCl, and 35 mM imidazole, and eluted with 20 mM sodium phosphate, pH = 7.4, 150 mM NaCl, and 200 mM imidazole. The eluted fraction was concentrated using the Amicon^®^ Ultra centrifugal filter with a 30 kDa cut-off membrane (Millipore, Darmstadt, Germany) and further purified in ÄKTA Pure System using a Superdex^®^ 200 10/300 GL column (GE Healthcare, Chicago, IL, USA) equilibrated in 20 mM sodium phosphate, pH = 7.4, and 150 mM NaCl.

### 2.3. CryoEM Grid Preparation and Data Collection

For cryo-EM grid preparation, 3 μL of purified YFV NS1 proteins was applied to the amorphous alloy film (CryoMatrix Au300-R12/13, Zhenjiang Lehua Electronic Technology Co., Ltd., Zhenjiang, China) that was glow-discharged at 15 W for 45 s. The grids were blotted for 3.5 s at 4 °C with 100% humidity using Vitrobot IV (Thermo Scientific, Winster, MA, USA), frozen in liquid ethane, and stored in liquid nitrogen.

The grids containing YFV NS1 proteins were imaged using a 300 kV Titan Krios Gi3 (USA) equipped with a Gatan K3 Summit detector. The movie stacks with 50 frames were automatically collected using SerialEM 4.1 software (https://bio3d.colorado.edu/SerialEM/, accessed on 17 June 2024) at a nominal magnification of 105,000×, corresponding to a pixel size of 0.85 Å. The defocus ranges from −1.2 to −1.8 μm. Each movie stack was exposed in the counted-nanoprobe mode for 1.96–2.28 s, and the total dose was approximately 50–54 e^−^/Å^2^. In total, 2702 movie stacks were collected with conventional imaging acquisition setting without tilting the stage. To address the issue of preferred orientation, a dataset of 2053 movie stacks was collected with the stage tilted at 15°.

### 2.4. CryoEM Data Processing of YFV NS1

For the non-tilting data, the movie frames were aligned using MotionCorr2 in RELION 4.0 (https://relion.readthedocs.io/en/release-4.0/, accessed on 17 June 2024), and the contrast transfer function (CTF) was estimated using Patch CTF in CryoSPARC v4.5 (https://cryosparc.com/, accessed on 17 June 2024). Overall, 2,788,529 particles were picked by template and extracted in a pixel size of 1.7 Å. Two-dimensional classifications and three-dimensional classifications were performed to remove junk particles, and 242,649 particles were retained. Ab initio model reconstruction generated multiple dimer and tetramer models. Then, particles comprising a dimer and tetramer were separated by heterogeneous refinement. The same data processing was applied to the tilted data, resulting in a number of 384,086 good particles. The particle set was merged with the not-tilting data for a round of 3D classifications. In total, 215,543 particles with good 3D alignment were selected for non-uniform refinement, and a map at 3.57 Å resolution was generated. The local refinement was performed by applying a mask on one of the dimers and yielded a map at 3.31 Å resolution. The orientation distribution and local resolution map were generated from cryoSPARC. Three-Dimensional Variability Analysis was performed in cryoSPARC with the final particle stack.

### 2.5. Model Building and Refinement

The model of YFV NS1 dimer was initially generated through homology modeling in SWISS-MODEL (https://swissmodel.expasy.org/, accessed on 17 June 2024) using the structure of ZIKV NS1 dimer (PDB: 5GS6) as template. One or two copies of the dimer models were fitted into the density maps comprising a YFV-focused refinement map and a YFV tetramer map, respectively. The chain IDs were changed, and the chains were merged in Coot (https://www2.mrc-lmb.cam.ac.uk/coot/coot/, accessed on 17 June 2024) to obtain the oligomer models. After one round of rigid body refinement, one round of morphing, and one round of simulated annealing in Phenix Real_Space_Refinement (https://phenix-online.org/, accessed on 17 June 2024), residues missing were manually built in Coot. Then, the models were refined by iterative refinement in Phenix Real_Space_Refinement with default parameters and manual adjustment in Coot. Secondary structure restrains were used for the real space refinement in Phenix. The geometries of refined models were validated using MolProbity (http://molprobity.biochem.duke.edu/, accessed on 17 June 2024). The figures of the structures were prepared in ChimeraX (https://www.rbvi.ucsf.edu/chimerax/, accessed on 17 June 2024) and PyMOL(https://www.pymol.org/, accessed on 17 June 2024).

## 3. Results

### 3.1. CryoEM Structures Analysis of YFV sNS1 Oligomers

Recombinant YFV NS1 proteins (residues 1 to 352) were expressed in 293F cells and purified by nickel affinity chromatography and size-exclusive chromatography (SEC). Although the SDS-PAGE showed that the purified YFV sNS1 proteins as a single band, the retention volumes on the Superdex 200 Increase column ranged from 10.3 to 10.8 mL, indicating that the proteins form oligomers rather than dimers (Appendix A). For cryoEM analysis, the peak fraction of the purified YFV sNS1 samples was concentrated and applied to cryo-grids, and 2702 movies were collected from a 300 kV Titan Krios (Appendix A). The two-dimensional (2D) class averages indicated that the sample was a mixture of tetramer and hexamer (Appendix A). Since the tetramer dominates the particles, the 3D structure of the tetramer was generated after ab initio model generation. In order to address the issue of preferred orientation, 2053 movies stacks were collected by tilting the sample stage at 15°. The final reconstruction of 215,543 particles yielded a structure (named YFV_NS1_tetramer) at a 3.57 Å resolution (Figure 1A and Appendix A; Table 1). The local resolution map shows that one of the dimers is blurry with a worse resolution, indicating its flexibility. Then, we performed focused refinement by applying a mask on one of the dimers and improved the resolution to 3.31 Å (Appendix A; Table 1). The atomic models of the YFV_NS1_tetramer and focused refined dimer were built according to the density map (Figure 1A–D and Appendix A). The chains were named AaBb and Aa for NS1 tetramer and dimer, respectively. Most of the residues could be traced in the tetramer structure, but some residues in the wing domain loop region (109–128) and the β-ladder loop region (339–352) were missing due to density blurring.

The NS1 monomer has 352 amino acids, forms 2 α-helices and 20 β-strands, and contains 12 conserved cysteine residues that form 6 intramolecular disulfide bonds. The NS1 monomer consists of three distinct domains: a β-roll domain (residues 1–29, β1–2), a wing domain (residues 38–151, β4–7 and α1–2), and a β-ladder domain (residues 181–352, β10–20) (Appendix A). β3, β8, and β9 are from a ‘‘connector’’ domain (residues 30–37 and 152–180) and connect the wing to the β-roll and β-ladder domains (Figure 1D). Loopβ8–β9 in the connector domain are also known as the greasy fingers and wing tips (Figure 1 and Appendix A). Generally, all flavivirus NS1 dimers have a cross shape, formed by the intertwining of Monomer A and a, showing two distinct surfaces [29] (Figure 1D and Appendix A). The β-roll domain, the greasy finger (loopβ8–β9), and loopβ5–β6 in the wing periphery region form the ‘inner’ hydrophobic face, while the wing domain and β-ladder form the outer surface (Figure 1D). By superimposing our NS1 dimer structure on the crystal structure of NS1c (PDB: 5YXA), the root mean square deviation (RMSD) is 0.69 Å (Appendix A), indicating that the overall conformation of the NS1 dimer is very stable, except for some conformational differences in the peripheral loop 337–352. In addition, we also use the Alpha fold to generate the 3D model of the YFV NS1 dimer (Appendix A). The structural comparison showed that the RMSD of the two dimers structures is 0.847 Å, suggesting the modelled dimer structure is very similar to that of the YFV_NS1 dimer.

### 3.2. The Assembly of YFV sNS1 Tetramer

The structure of the YFV_NS1_tetramer presents an asymmetrical “H” shape, which consists of two NS1 dimers (Aa and Bb) stacked through the inner hydrophobic surfaces (Figure 1A–C). The distance between the centroids of Monomer A and B (42.5 Å) is smaller than the distance of Monomer a and b (54.6 Å) (Figure 2A). Thus, Monomer B leans towards Monomer A, while Monomer b extends outward. The buried interface area between dimer Aa and Bb is 853.7 Å^2^ via a PISA calculation [30]. The tetrameric structure showed that the interface between dimer Aa and Bb is mainly contributed by the hydrophobic residues from the β-roll domains and greasy fingers (Figure 2B, Appendix A). To describe the interface, a central stacking zone (consisting of the β-roll domains) and an upper stacking zone (consisting of the β-roll domains and the greasy fingers) are defined, respectively (Figure 2C). In the central stacking zone, the β2a-β1a-β1A-β2A plane packs against the opposite β2b-β1b-β1B-β2B plane with a dislocation. As shown in Appendix A, the residues at the interface are mainly hydrophobic. In the upper stacking zone, Phe163A in loopβ8-β9A interacts with Gln2b and Gly16b in loopβ1-β2b, while Val162B in loopβ8-β9A interacts with Phe8A in loopβ1A (Figure 2D and Appendix A). Therefore, the YFV NS1 tetramer is asymmetrically stacked with two dimers, and the interface is mainly mediated by the β-roll and the greasy fingers.

### 3.3. The YFV Tetramer Is a Semi-Stable Structure

Shu et al. previously reported that the recombinant DENV2 sNS1 tetramers mainly occupy two oligomeric states: stable and loose conformations [23]. To understand the oligomerization of flavivirus sNS1, we compared our structures with the cryoEM structures of DENV2 tetramers (7WUT, stable tetramer and 7WUU loose tetramer) and crystal packing of ZIKV sNS1 tetramers (PDB: 5GS6). Since the DENV2 loose tetramer was in a poor resolution, Shu et al. deposited the cryoEM map and a model of dimer (PDB: 7WUS). We fitted two copies of dimers in the the cryoEM map (EMDB: 32842) and generated the model of the DENV2 loose tetramer for comparison, which is labeled as 7WUU in Figure 3A and Appendix A.

For comparison, the models of dimer Aa were aligned, and the differences of dimer Bb are illustrated in Figure 3A–C. Firstly, the superimposition our YFV sNS1 tetramer with the DENV2 and ZIKV loose tetramers showed that they have large conformational differences (Figure 3). Although the package of the core β-roll domain of these three tetramers is very similar, the wing domain of Monomer B shifted towards Monomer A in two loose tetramer models (Figure 3A,B). However, the overall structure of dimer Bb shifted towards dimer Aa of the DENV2 stable tetramer model, compared to our YFV tetramer model. Furthermore, the elongated β sheets of the N-terminal domain specific to the stable tetramer were not found in our tetramer electron microscope map. These results suggest that our YFV sNS1 tetramer may be neither a stable nor a loose tetramer. In our YFV tetramer, the distance between the centroids of Monomers A and B is larger than in the loose tetramers of DENV2 and ZIKV (42.5 Å vs. 35.8 Å and 37.6 Å), whereas the distance between Monomers a and b is smaller (54.6 Å vs. 59.7 Å and 57.0 Å) (Figure 3D). The stable tetramer of DENV2 has a more compact assembly between dimer Aa and Bb, where both have a distance of 37.0 Å. The buried interface area between dimer Aa and Bb of YFV is 853.7 Å^2^ via a PISA calculation. However, the DENV2 and ZIKV tetramers have a buried interface of ~650–690 Å^2^ [23], which is much smaller than the one we observed in the YFV tetramer. These observations also indicate that the YFV tetramer is in a conformation that is distinct from the loose tetramer and stable tetramer. Therefore, we define this structure as a “semi-stable” tetramer, which may represent the intermediated state between the stable tetramer and the loose tetramer.

### 3.4. The Dynamic Characteristics of YFV NS1 Tetramer

According to the comparison among the tetrameric flavivirus NS1, we consider that the NS1 dimer has a semi-stable structure in the solvent. We also noticed that the local resolution of dimer Bb is worse than dimer Aa, indicating that dimer Bb may occupy multiple conformations. To understand the flexibility of the tetramer, we performed 3D Variability Analysis in CryoSPARC. Unsurprisingly, we captured two discrete maps in different conformations, named state1 and state2 (Figure 4A,B, and Appendix A). Due to the resolution limitation, the models of the two dimers were fitted into state1 and state2 maps (Figure 4A,B). By superposing the structure of Aa, we compared the conformations of dimers Bb between state1 and state2. The RMSD of the dimer Bb of the two states of tetramers is 2.785 Å. The distances of centroid between Monomer A-B and a-b were measured. Compared with state1, the distance of centroid A-B from state2 is larger (42.1 A to 44.3 A), while the distance of centroid a-b is smaller (56.4 A to 54.3 A). The core stacking zone of state2 has a subtle shift, resulting in an outward tilt of the wing domain of Monomer B and an inward tilt of Monomer b (Figure 4C). In order to better characterize the relative movement of dimer Bb, we also defined the dimer axis by linking the centroids of two monomers in a dimer and measured the tile angle of the dimer Bb. The centroid of the dimer Bb of state 2 is tilted by 6.7° relative to the structure of state 1. These results suggest that the structure of YFV_NS1_tetramer is flexible with dynamic conformations (Appendix A).

## 4. Discussion

In this study, we investigated the oligomeric structures of recombinant YFV sNS1 dimers using cryoEM. Two-dimensional averages showed that recombinant YFV sNS1 is predominantly a tetramer with fewer hexameric particles. These results are consistent with the observations reported by Syzdykova et al. that GFP-NS1 fusion proteins from YFV-infected cells may contain tetramers and hexamers [26]. Although the proportions of the NS1 tetramer and hexamer may differ from our study, we speculate that this may be influenced by the differences in expressing cell line and purification conditions. To illustrate the oligomerization of sNS1, we also determined the cryoEM structures of the tetramer at a resolution of 3.57 Å. The YFV NS1 tetramer has an asymmetrical “H” shape, where the top two monomers are more compact and the bottom two are more loosely stacked. In contrast, the stable tetramer of DENV2 NS1, as reported by Shu et al., presents a symmetrical structure [23]. This discrepancy may be because we did not impose any symmetry constraints during the reconstruction of the YFV tetramer, whereas a D2 symmetry was imposed during the reconstruction of the DENV2 stable tetramer. In addition, the DENV2 stable tetramers showed elongated β-sheets at the interface of the two dimers, which was not observed in the YFV tetramer structure.

The assembly for the structure of the YFV_NS1_tetramer is facilitated by a cluster of hydrophobic interactions between the β-roll domain and greasy fingers, which are similar to those of the crystal packing process of the ZIKV NS1 loose tetramer [29]. However, in the YFV_NS1_tetramer, the upper two monomers are positioned further apart, while the lower two are closer together, resulting in a larger interface area between the two dimers. Altogether, the structure of the YFV_NS1_tetramer is defined as a semi-stable tetramer. Moreover, we also found the YFV tetramer to have multiple conformations, suggesting that the assembly of the YFV tetramer is flexible and that it has the potential to shift between stable tetramer and loose tetramer states.

Although the flavivirus NS1 dimer has been extensively studied in vitro, its oligomeric state in the physiological environment remains unclear. Recently, Harris Group reported a significant correlation between the serum levels of YFV sNS1 and clinical laboratory parameters, such as viral load and disease severity [15]. The physiological significance of the different oligomeric forms of sNS1 may be more complicated and requires further investigation. These results improve our understanding of the oligomerization of sNS1 and provide valuable insights into the development of NS1-based diagnostic tools and therapeutic strategies.

## Figures and Tables

**Figure 1 viruses-16-01212-f001:**
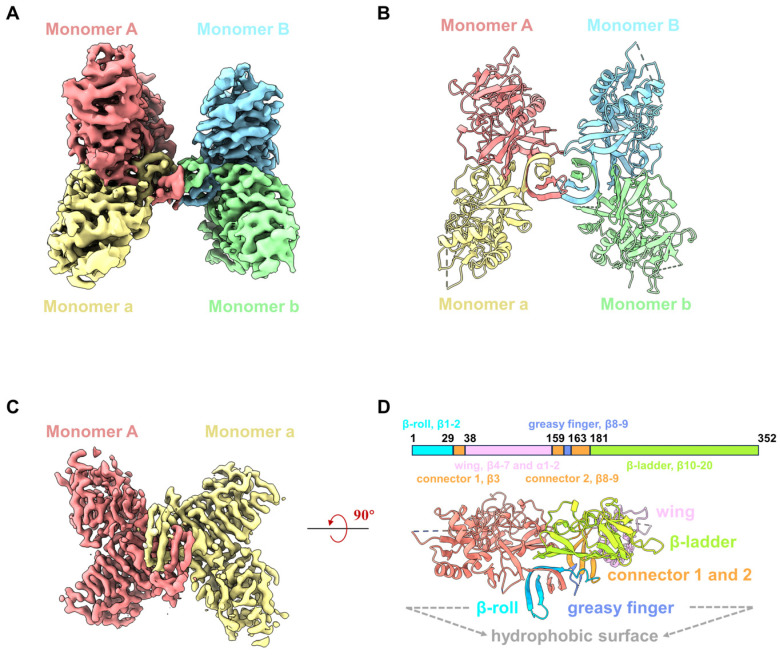
CryoEM structures of YFV sNS1. (**A**) Cryo-EM density map of YFV NS1 tetramer (threshold lever: 0.22). (**B**) Atomic model of YFV NS1 tetramer in cartoon representation. (**C**) Cryo-EM density map of YFV NS1 focused refined dimer (threshold lever: 0.353). (**D**) Schematic representation of YFV NS1 and cartoon model of YFV NS1 dimer. In (**A**–**D**), density maps and atomic models are colored by chains in the same color scheme: Monomer A: salmon; Monomer a: khaki; Monomer B: light sky blue; Monomer b: light green.

**Figure 2 viruses-16-01212-f002:**
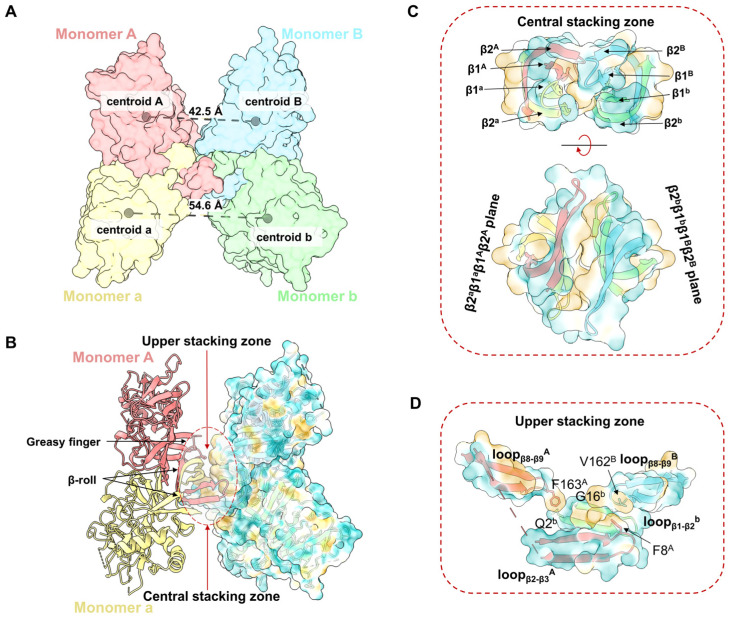
Assembly of YFV sNS1 tetramer. (**A**) Centroidal distance between dimer Aa and dimer Bb. The model of YFV NS1 tetramer is colored by chains and shown as surface. The centroids and corresponding distances are labeled in gray spheres and dashed lines. (**B**) Stacking zone in YFV NS1 tetramer. The YFV NS1 tetramer is presented in image. Surface model of Monomers B and b are presented with 80% transparency and colored by hydrophobicity. Upper and central stacking zones are highlighted in circles. (**C**) Interactions in the central stacking zone. Both side view (top) and top view (bottom) are presented. (**D**) Interactions in the upper stacking zone. Key residues (F8^A^, F163^A^, V162^B^, Q2^b^, G16^b^, analyzed by LigPlot+) are shown in sticks [31]. In (**B**–**D**), the cartoon models are colored by chains, and the surface models are colored by hydrophobicity in ChimeraX with 80% transparency.

**Figure 3 viruses-16-01212-f003:**
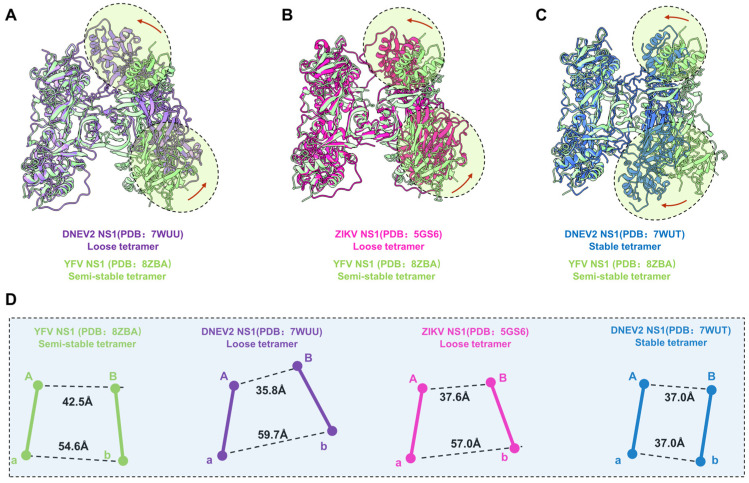
Comparison of YFV, ZIKV, and DENV2 NS1 tetramer models. (**A**–**C**) Atomic models of DENV2 NS1 loose tetramer, ZIKV NS1 loose tetramer, and DENV2 NS1 stable tetramer aligned to YFV NS1 semi-stable tetramer. Domain switch directions are highlighted in circles and indicated by red arrows. (**D**) Centroidal distance between dimer Aa and dimer Bb. The Centroid distances between Monomer A and B, and between Monomer a and b are measured for each of the structure presented in (**A**–**C**). Images of each NS1 and the corresponding centroids are colored, as follows: YFV NS1 semi-stable tetramer: light green; DENV2 NS1 loose tetramer: medium purple; ZIKV NS1 loose tetramer: deep pink; DENV2 NS1 stable tetramer: royal blue.

**Figure 4 viruses-16-01212-f004:**
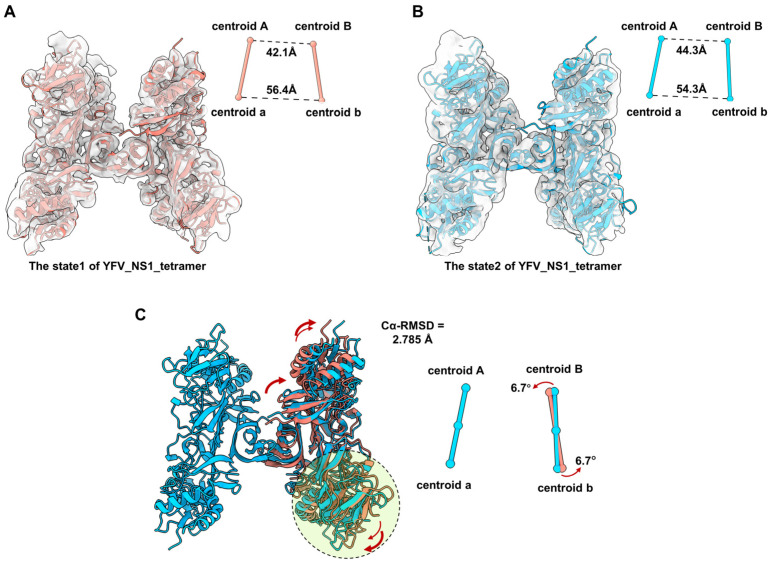
Comparison of the conformational change between two states of YFV_tetramer_NS1 models. (**A**,**B**) Atomic models of YFV NS1 tetramer in state1 and state2 (threshold levels: 0.128 and 0.108, respectively). (**C**) Comparation between the models of YFV_NS1_tetramer_state1 and YFV_NS1_tetramer_state 2. In (**A**–**C**), the centroids and dimer axes for each tetramer are shown near the atomic models in spheres and sticks. The centroidal distances are labeled in black dashed lines. The tilt directions and angles are labeled in red arrows. Models of each NS1 and the corresponding centroids and axes are colored, as follows: YFV_NS1_tetramer_state1: salmon; YFV_NS1_tetramer_state2: dodger blue.

**Table 1 viruses-16-01212-t001:** Statics of data collection, data process, model refinement, and validation.

	YFV_NS1 Focused Dimer (EMDB: 39898) (PDB: 8ZB9)	YFV_NS1_tetramer (EMDB: 39899) (PDB: 8ZBA)
Data and processing	v	
Magnification	105,000	105,000
Voltage (kV)	300	300
Electron exposure (e^−^/Å^2^)	50	50
Defocus range (μm)	−1.2~−1.8	−1.2~−1.8
Pixel size (Å)	0.85	0.85
Symmetry imposed	C1	C1
Initial particle images (no.)	4,793,330	4,793,330
Final particle images (no.)	215,543	215,543
Map resolution (Å)	3.31	3.57
FSC threshold	0.143	0.143
Map resolution range	3.0–4.5	3.0–5.0
Refinement		
Initial model used (PDB code)	-	-
Map sharpening B factor (Å^2^)	−172.8	−183.0
Model composition		
Non-hydrogen atoms	4912	9804
Protein residues	623	1244
Ligands	-	-
B factor (Å^2^)		
Protein	54.32	54.32
Ligand	-	-
R.m.s. deviations		
Bond lengths (Å)	0.006	0.007
Bond angles (°)	0.830	0.893
Validation		
MolProbity score	2.33	2.40
Clash score	14.87	7.10
Rotamers outliers (%)	0.00	0.92
Ramachandra plot		
Favored	85.50	89.28
Allowed	14.33	14.98
Outliers	0.16	0.00
Model-map FSC (0.143)	3.43	3.78

## Data Availability

The authors confirm that the data supporting the findings of this study are available within this article.

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
