# Peer review of "Structural Insights into the Dynamic Assembly of a YFV sNS1 Tetramer"

_viruses, 2024, doi:10.3390/v16081212_

Round 1

Reviewer 1 Report

Comments and Suggestions for Authors

This manuscript describes the 3D structure of the soluble (secreted) form of the Yellow Fever Virus NS1 protein. Purified, recombinant NS1 secreted from 293 cells is purified and analyzed by cryo-electron microscopy. The 3D reconstructions, refined to 3.31 Å, indicate that most of the recombinant, secreted NS1 protein is found as "H-shaped" tetramers, composed of two dimers interacting with their hydrophobic surfaces. Moreover, tetramers are suggested to be flexible, with one dimer moving about the other almost 7 degrees. This manuscript helps understand the oligomeric states of the secreted form of mosquito-borne flaviviruses NS1, a protein related to pathogenesis in several ways. The manuscript is well written and the data supports the author's conclusions.  

Comments:

1. Since no 3D cristal of the full NS1 protein of YFV is available (only a carboxy-terminal fragment), authors should try to fit into their 3D models, structural models generated with Alpha-fold, and include the result as an additional supplemental figure. 

2. Please include in the manuscript information about the percentages of hexamers versus tetramers versus dimers that the authors observed in their 2D preparations. This information will interest the field because the hexamer is supposed to be the "classic" secreted form of dengue NS1. 

3. Authors may be interested in the following article: Chew, B.L.A., Ngoh, A.Q., Phoo, W.W. et al. Structural basis of Zika virus NS1 multimerization and human antibody recognition. npj Viruses 2, 14 (2024).

Minor comments:

1. Use italics to refer to the Flavivirus genus and Flaviviridade family.

2. Page 1, lane 39. Replace the word "mysterious" for multifunctional.  

Comments on the Quality of English Language

The English language requires minor revision, especially the use of capital letters.

Author Response

This manuscript describes the 3D structure of the soluble (secreted) form of the Yellow Fever Virus NS1 protein. Purified, recombinant NS1 secreted from 293 cells is purified and analyzed by cryo-electron microscopy. The 3D reconstructions, refined to 3.31 Å, indicate that most of the recombinant, secreted NS1 protein is found as "H-shaped" tetramers, composed of two dimers interacting with their hydrophobic surfaces. Moreover, tetramers are suggested to be flexible, with one dimer moving about the other almost 7 degrees. This manuscript helps understand the oligomeric states of the secreted form of mosquito-borne flaviviruses NS1, a protein related to pathogenesis in several ways. The manuscript is well written and the data supports the author's conclusions.

Comments:

1. Since no 3D cristal of the full NS1 protein of YFV is available (only a carboxy-terminal fragment), authors should try to fit into their 3D models, structural models generated with Alpha-fold, and include the result as an additional supplemental figure.

Response 1: Thank you for the insightful suggestions, we agree with this comment. We have added the description of YFV NS1 dimer 3D models generated with Alpha-fold in the Results section in the revised manuscript. The structural comparison showed that the RMSD of two dimers structures is 0.847 Å, suggesting the modelled dimer structure is very similar to that of YFV_NS1 fouced dimer (page 5, line 183 to 185).

2. Please include in the manuscript information about the percentages of hexamers versus tetramers versus dimers that the authors observed in their 2D preparations. This information will interest the field because the hexamer is supposed to be the "classic" secreted form of dengue NS1.

Response 2: Thank you for the constructive comments. The 2D averages results showed that recombinant YFV sNS1 is predominantly tetramer with fewer hexameric particles. We have added the percentages of hexamers versus tetramers that the observed in their 2D preparations in the Supplementary Materials section in the revised manuscript.

3. Authors may be interested in the following article: Chew, B.L.A., Ngoh, A.Q., Phoo, W.W. et al. Structural basis of Zika virus NS1 multimerization and human antibody recognition. npj Viruses 2, 14 (2024).

Response 2: Thank you for the constructive comments. We have added the citation in the revised manuscript.

Minor comments:

Point 1: Use italics to refer to the Flavivirus genus and Flaviviridade family.

Response 1: Thank you for the constructive comments. We have corrected the contents in the revised manuscript (page 1, line 33).

Point 2: Page 1, lane 39. Replace the word "mysterious" for multifunctional.

Response 2: Thank you for the constructive comments. We have corrected the contents in the revised manuscript (page 1, line 39).

Reviewer 2 Report

Comments and Suggestions for Authors

The paper describes the tetrameric nature of the Yellow fever virus (YFV) NS1 protein, supported by two CryoEM structures at 3.5 and 3.3 Å resolution. Paper is very clearly written and beautifully illustrated, contains an UpToDate literature review and very detailed methods section. The main part of the paper focuses on the structural description of the interaction between 2 YFV NS1dimers within the tetramer and interface flexibility between NS1 proteins from different sources. However, the connection between the oligomeric state of the protein and its benefits for NS1-based diagnosis and therapies wasn't very clear.

Minor comments:

  • Please include CryoEM map level(s) to figure captions with maps contoured around the structures.
  • Please include model vs map FSC numbers to the table for both models.
  • Could you please provide examples of the CryoEM density, at the interdimer interface and around one of the most defined beta strands?

Author Response

The paper describes the tetrameric nature of the Yellow fever virus (YFV) NS1 protein, supported by two CryoEM structures at 3.5 and 3.3 Å resolution. Paper is very clearly written and beautifully illustrated, contains an UpToDate literature review and very detailed methods section. The main part of the paper focuses on the structural description of the interaction between 2 YFV NS1dimers within the tetramer and interface flexibility between NS1 proteins from different sources. However, the connection between the oligomeric state of the protein and its benefits for NS1-based diagnosis and therapies wasn't very clear.Based on the maps presented by the authors, the “dimer” class seems to be merely a tetramer with a stable dimer with another flexible dimer, as evidenced by 2D classes (Fig. S1B, row1, column 4) and fuzzy density in cryo-EM reconstruction. Thus, the authors should refrain from referring this as a “dimer”, but rather a more flexible tetramer with dimer resolved, etc, unless there is more concrete evidence showing there is true dimer present in cryo-EM dataset, or independent methods validating the presence of a dimer population in sample.

Minor comments:

1. Please include CryoEM map level(s) to figure captions with maps contoured around the structures.

Response 1: Thank you for the insightful suggestions. We have provided threshold lever values in figure 1 and 4 figure legends to show local map quality and map-model fit quality in the revised manuscript.

2. Please include model vs map FSC numbers to the table for both models.

Response 2: Thank you for the constructive comments. We have added model vs map FSC numbers in the revised manuscript (page 5) (Table 1).

Table 1. Statics of data collection, data process, model refinement, and validation.

YFV_NS1

focused dimer

(EMDB: 39898)

(PDB: 8ZB9)

YFV_NS1_tetramer

(EMDB: 39899)

(PDB: 8ZBA)

Data and processing

Magnification

105,000

105,000

Voltage (kV)

300

300

Electron exposure (e-/ Å2)

50

50

Defocus range (μm)

-1.2 ~ -1.8

-1.2 ~ -1.8

Pixel size (Å)

0.85

0.85

Symmetry imposed

C1

C1

Initial particle images (no.)

4,793,330

4,793,330

Final particle images (no.)

215,543

215,543

Map resolution (Å)

3.31

3.57

FSC threshold

0.143

0.143

Map resolution range

3.0 – 4.5

3.0 - 5.0

Refinement

Initial model used (PDB code)

-

-

Map sharpening B factor (Å2)

-172.8

-183.0

Model composition

Non-hydrogen atoms

4912

9804

Protein residues

623

1244

Ligands

-

-

B factor (Å2)

Protein

54.32

54.32

Ligand

-

-

R.m.s. deviations

Bond lengths (Å)

0.006

0.007

Bond angles (°)

0.830

0.893

Validation

MolProbity score

2.33

2.40

Clash score

14.87

7.10

Rotamers outliers (%)

0.00

0.92

Ramachandra plot

Favored

85.50

89.28

Allowed

14.33

14.98

Outliers

0.16

0.00

Model-map FSC (0.143)

3.43

3.78

  1. Could you please provide examples of the CryoEM density, at the interdimer interface and around one of the most defined beta strands?

Response 3: Thank you for the insightful suggestions, we agree with this comment. We have

provided an additional supplemental figure 4 to show local map quality and map-model fit quality in the revised manuscript.

Figure S4. The local density maps of YFV_NS1_tetramer and YFV_NS1 focused dimer

(A)The local density map of the β-roll domains of dimer Aa from YFV_NS1_tetramer. (B) The local density map of the β-roll domains of dimer Bb from YFV_NS1_tetramer. (C) The local den-sity map of the greasy finger of monomer A from YFV_NS1_tetramer. (D) The local density map of the greasy finger of monomer B from YFV_NS1_tetramer. (E) The local density map of the β-roll domains of dimer Aa from YFV_NS1 focused dimer. (F) The local density map of the greasy finger of dimer Aa from YFV_NS1 focused dimer.
